# Long-Wave Infrared Polarization-Based Airborne Marine Oil Spill Detection and Identification Technology

Hongyu Sun, Lianji Ma, Qiang Fu *, Yingchao Li, Haodong Shi *, Zhuang Liu, Jianan Liu, Jiayu Wang and Huilin Jiang

National and Local Joint Engineering Research Center of Space Optoelectronics Technology, Changchun University of Science and Technology, Changchun 130022, China
* Correspondence: fuqiang@cust.edu.cn (Q.F.); shihaodong@cust.edu.cn (H.S.)

**Abstract:** In this paper, infrared polarization detection information acquisition technology is proposed, and the polarization characteristics of oil spills are modeled and studied. A set of long-wave infrared polarization detection equipment for oil spills is designed and built, and modeling research on oil spill polarization characteristics is carried out to accurately detect and identify oil spill types and for the faster processing of oil spill events. Oil spill accuracy is increased by defining the polarization maintenance method of the polarization optical system and reducing the polarization measurement error brought on by the imaging system. As a result, a higher than 3% contrast exists between the polarization degree image and the corrected infrared intensity image. Outdoor tests using oil, palm oil, crude oil, gasoline, and diesel oil spill types are carried out in a controlled environment to collect data on the polarization of various oil species. According to the findings, each oil species' infrared polarization contrast with seawater is typically greater than its infrared intensity contrast. However, the polarization data of saltwater, diesel, and palm oil, which are difficult to identify in intensity data, show a noticeable difference, further proving the viability of utilizing polarization to discern oil spills.

**Keywords:** polarization; oil spill; photoelectric detection; long-wave infrared; oil species differentiation





## 1. Introduction

Oil resources are vital to support the sustainable development of national economic industries. In recent years, multiple underwater pipeline ruptures caused by ecological catastrophes have shaken the world [1,2]. The growth of the petroleum business not only strengthens the global economy but also enhances the standing of the global oil industry on the worldwide market. The International Tanker Owners Pollution Federation (TIOPF) estimates that tanker accidents cost the globe 5.86 million tons of oil [3,4]. In the event of an oil leak during the storage and transportation of oil, it is crucial to find and identify the oil species as quickly as possible. Currently, synthetic aperture radar (SAR) satellite remote sensing monitoring, aerial remote sensing monitoring, ship remote sensing monitoring, spot monitoring, and buoy tracking—among which Yin is one—are the primary technical techniques for locating offshore oil spills [5,6]. De Carolis et al. used near-infrared MERIS and MODIS images to estimate the thickness of the ocean oil slick [7,8], and in 2010 Dayi et al. used a UV push-sweep imaging remote sensing method to monitor real oil spill pollution on the sea surface [9,10]. Furthermore, Lacava et al. used data from the visible channel of the Moderate Resolution Imaging Spectroradiometer (MODIS) [11,12]. Therefore, our team studied how the oil spill affected infrared polarization, and the photoelectric conversion process allowed us to assess the sources of various errors and calibrate the error correction. Bias preservation is accomplished by inverting the Mueller matrix with the information from the outgoing light. Outdoor experiments were conducted using the general design of an airborne platform-based marine monitoring multi-dimensional high-resolution optical imaging system. The polarization of the reflected light of each

sample was measured for gasoline, fuel oil, diesel oil, palm oil, crude oil, and seawater in descending order. The corresponding polarization 95% confidence interval did not intersect, indicating that the polarization can be used to distinguish oil species. In the complex marine environment interference, compared with the traditional intensity detection information, the polarization information is less affected, which can highlight the characteristics of an oil spill, and is conducive to the effective identification and accurate differentiation of targets. Compared with other experiments that use polarized light to detect oil flowing in the sea, we study the causes of errors caused by receiving polarized light, which is of great significance to ensure the accuracy of polarization state inversion of incident light. Moreover, the experiment of maintaining bias effect is carried out, which provides a reliable theoretical basis for the experiment of oil species differentiation.

In this paper, infrared polarization detection information acquisition technology is proposed, and the polarization characteristics of oil spills are modeled and studied. Finally, a set of long-wave infrared (electromagnetic waves with wavelengths of 8 to 12 μM) polarization detection equipment for oil spills is designed and built.

## 2. Basic Theory

### 2.1. Modeling the Infrared Polarization Characteristics of Oil Spills

In 1852, Stokes realized the construction of completely polarized state, unpolarized state and partially polarized state model only through a $4 \times 1$ matrix, namely, Stokes vector, whose expression is as follows [13,14]:

$$\text{Stokes} = \begin{bmatrix} S_0 \\ S_1 \\ S_2 \\ S_3 \end{bmatrix} = \begin{bmatrix} I_0 + I_{90} \\ I_0 - I_{90} \\ I_{45} - I_{135} \\ I_R - I_L \end{bmatrix}. \tag{1}$$

In the above formula, $S_0$ is the total intensity of light, $S_1$ is the intensity difference between the polarization directions of $0°$ and $90°$, $S_2$ is the intensity difference between the polarization directions of $45°$ and $135°$, and $S_3$ is the intensity difference between right-handed and left-handed circularly polarized light.

The DoP (Degree of Polarization) is expressed by Stokes vector as [15,16]:

$$\text{DoP} = \frac{\sqrt{S_1^2 + S_2^2 + S_3^2}}{S_0}. \tag{2}$$

AoP (Angle of polarization) refers to the angle between the direction of optical vibration and the specified $0°$, which is defined as follows [17,18]:

$$\text{Aop} = \frac{1}{2}\arctan\left(\frac{S_2}{S_1}\right). \tag{3}$$

The oil spill's infrared polarization in the ocean backdrop is mostly made up of the polarization of the spill's radiation and the polarization of its reflection [19,20]. The radiance of the oil spill affects the polarization of the spontaneous radiation, and the reflectance affects the polarization of the reflected radiation. Therefore, the equation for infrared radiation can be written as

$$I_P(\varphi, \lambda, T, \theta, N) = I_{w,P}(T, \lambda) + R_P(\theta, \lambda, N)I_b(\theta, \lambda), \tag{4}$$

$$I_s(\varphi, \lambda, T, \theta, n) = I_{w,s}(T, \lambda) + R_s(\theta, \lambda, n)I_b(\theta, \lambda). \tag{5}$$

The element information in the above formula is shown in Table 1, and Formulas (4) and (5) are illustrated in Figure 1.

**Table 1.** The meaning of the symbols in Formulas (4) and (5).

| Symbol | Meaning |
|---|---|
| $I_p$ | p component light intensity |
| $I_s$ | s component light intensity |
| $I_{wp}$ | The P component of the object's own radiation |
| $I_{ws}$ | The s component of the object's own radiation |
| $R_p$ | The reflectance of the p component |
| $R_s$ | The reflectance of the s component |
| $I_b$ | Background radiation intensity |
| $\varphi$ | Radiation angle |
| $\lambda$ | Wavelength |
| T | Temperature |
| $\theta$ | Angle of incidence |
| N | The refractive index of the medium of p component |
| n | The refractive index of the medium of s component |

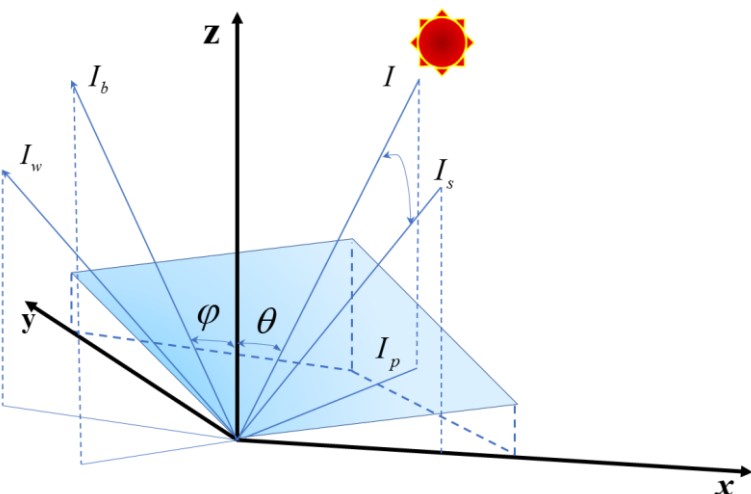

**Figure 1.** The element information in Formulas (4) and (5).

Wien's displacement law reveals the relationship between the peak of spectral emission of the blackbody, the corresponding wavelength of radiation peak, and the blackbody temperature, expressed as [21]:

$$\lambda_m T = b, \tag{6}$$

where the constant $b$ has the value $2898.8 \pm 0.4$ (μM K), according to Wien's displacement equation, and the highest radiation wavelength from objects at ambient temperature is mainly in the infrared long-wave band. In contrast, the peak wavelength of light from the sun is in the visible band.

According to Kirchhoff's law in the theory of thermal radiation, the ratio of a target's radiant emission of light at $\lambda$ wavelength $M^\lambda$ to its emissivity $\varepsilon$ is constant and equal to the radiant emission of a blackbody at that temperature $M_{bb}^\lambda$ when the target temperature is constant.

$$\frac{M^\lambda}{\varepsilon} = M_{bb}^\lambda, \tag{7}$$

$$I_{bb}^\lambda = \int_s M_{bb}^\lambda. \tag{8}$$

When the temperature is fixed, it can be calculated by Planck's radiation law.

$$M_{bb}^\lambda = \frac{c_1}{\lambda^5} \cdot \frac{1}{e^{c_2/(\lambda T)} - 1}, \tag{9}$$

where $\lambda$ is the wavelength ($\mu$M); $T$ is the absolute temperature (K); $c_1 = (3.7415 \pm 0.0003) \times 10^8$ (W $\cdot$ $\mu$m/m$^2$); $c_2 = (1.43879 \pm 0.00019) \times 10^4$ ($\mu$m/K).

In order to obtain the total intensity of the blackbody with area S in the long-wave infrared band, the above formula is integrated to obtain the following:

$$I_{bb}^{\lambda_1 - \lambda_2} = \int_{\lambda_1}^{\lambda_2} \int \frac{c_1}{\lambda^5} \cdot \frac{1}{e^{c_2/(\lambda T)} - 1} dS d\lambda. \tag{10}$$

Fresnel's law is as follows [22]:

$$r_p = \frac{E_{rp}}{E_{ip}} = \frac{n_2 \cos\theta_i - n_1 \cos\theta_t}{n_2 \cos\theta_i + n_1 \cos\theta_t} = \frac{\tan(\theta_i - \theta_t)}{\tan(\theta_i + \theta_t)}, \tag{11}$$

$$r_s = \frac{E_{rs}}{E_{is}} = \frac{n_1 \cos\theta_i - n_2 \cos\theta_t}{n_1 \cos\theta_i + n_2 \cos\theta_t} = \frac{\sin(\theta_i - \theta_t)}{\sin(\theta_i + \theta_t)}. \tag{12}$$

The reflectance of the $p$, $s$ component can be derived from Fresnel's law as follows:

$$R_s = \frac{(a - \cos\theta)^2 + b^2}{(a + \cos\theta)^2 + b^2}, \tag{13}$$

$$R_p = R_s \frac{(a - \sin\theta \tan\theta) + b^2}{(a + \sin\theta \tan\theta)^2 + b^2}, \tag{14}$$

where $a^2 = \left[ \sqrt{(n^2 - k^2 - \sin^2\theta)^2 + 4n^2k^2} + (n^2 - k^2 - \sin^2\theta) \right] / 2$; $b^2 = \left[ \sqrt{(n^2 - k^2 - \sin^2\theta)^2 + 4n^2k^2} - (n^2 - k^2 - \sin^2\theta) \right] / 2$ is the complex refractive index of the object, $n$ is the refractive index, and $k$ is the absorption coefficient.

The polarization of long-wave infrared radiation is

$$P = \frac{R_s(\theta, n, k) - R_p(\theta, n, k)}{2 - R_p(\theta, n, k) - R_s(\theta, n, k)}. \tag{15}$$

### 2.2. Study on Infrared Polarization Maintenance of Oil Spills

For offshore oil spill detection, this chapter studies the infrared polarization characteristics of oil spills on the ideal sea surface based on Fresnel theory, then builds a polarization model of the ideal sea surface based on infrared polarization theory, and carries out simulation analysis on the relationship between each influencing factor and polarization degree. Furthermore, to ensure the accuracy of infrared polarization information of the target in actual detection, the causes of the errors of a polarization camera with a micropolarizer array are analyzed. In addition, the methods of maintaining polarization are studied.

2.2.1. Infrared Polarization Error Analysis of Oil Spills

Several factors of error generation can be analyzed through the photoelectric conversion process [23,24], which are the polarization error generated by the characteristics of the optical system itself, the MPA (Micro Polarization Array) manufacturing error, and the error in the calibration and correction process. Each factor is analyzed below.

1.   The polarization characteristics of the optical system produce errors.

After the incident light passes through the polarization device, the Stokes vector changes accordingly. The relationship between the outgoing and incident light Stokes vectors can be represented by a Mueller matrix of order 4. The Mueller matrix at refraction can be expressed as [25,26]:

$$M_{tk} = \frac{\sin 2\theta_{tk} \sin 2\theta_{tk}}{2(\sin b_k \cos a_k)^2} \begin{bmatrix} \cos^2 a_k + 1 & \cos^2 a_k - 1 & 0 & 0 \\ \cos^2 a_k - 1 & \cos^2 a_k + 1 & 0 & 0 \\ 0 & 0 & -2\cos a_k & 0 \\ 0 & 0 & 0 & -2\cos a_k \end{bmatrix}, \quad (16)$$

where $\theta_{ik}$ is the angle of incidence for the $k$ refraction, $\theta_{ik}$ is the angle of refraction for the $k$ refraction, $a_k = \theta_{ik} - \theta_{tk}$, $b_k = \theta_{ik} + \theta_{tk}$.

By summarizing the polarization of visible $0°/90°$ linearly polarized light and $45°/135°$ linearly polarized light in the optical system, we can see that the number of light refractions in the system and the difference between the angle of the incident light and the outgoing light will affect the polarization of the outgoing light. Generally, a smaller number of refractions leads to a smaller change of polarization: the smaller the difference between the angle of the incident light and the outgoing light, the smaller the change of polarization, and the better the optical system preserves polarization. For partially polarized light, which can be understood as the result of the joint action of unpolarized and fully polarized light, the conclusion about the polarization degree still applies. This can provide the theoretical basis for designing a polarization optical system to preserve polarization.

2.     MPA manufacturing error.

The error of the polarization device itself is the leading cause of image non-uniformity [27,28], and the non-uniformity of the MPA imaging system is mainly determined by the material and level of the manufactured micropolarizers. After the micropolarizers are manufactured, the processing error causes the parameters of each micropolarizer to deviate from the theoretical value to different degrees, which eventually causes the imaging non-uniformity. The Mueller matrix of a single micropolarizer can be expressed as follows [29]:

$$M = \frac{1}{2} \begin{bmatrix} 1 & \frac{t_x - t_y}{t_x + t_y}\cos 2\theta & \frac{t_x - t_y}{t_x + t_y}\sin 2\theta & 0 \\ \frac{t_x - t_y}{t_x + t_y}\cos 2\theta & \cos^2 2\theta + 2\sqrt{t_x t_y}\sin^2 2\theta & \left(1 - 2\sqrt{t_x t_y}\right)\sin 2\theta \cos 2\theta & 0 \\ \frac{t_x - t_y}{t_x + t_y}\sin 2\theta & \left(1 - 2\sqrt{t_x t_y}\right)\sin 2\theta \cos 2\theta & \cos^2 2\theta + 2\sqrt{t_x t_y}\sin^2 2\theta & 0 \\ 0 & 0 & 0 & 0 \end{bmatrix}. \quad (17)$$

For $t_x, t_y, \theta$, if the maximum transmittance corresponds to the polarization direction of $x$ and the minimum transmittance corresponds to the direction of $y$, then the direction of $x$ is the direction of the transmittance axis $t_y$. The ratio of $t_x$. can characterize the performance of MPA, which is called the extinction ratio and is expressed by $\varepsilon$. When the light polarization direction is parallel to the $x$ axis, theoretically $t_x = 1, t_y = 0$, $\varepsilon = 0$, the light is completely transmitted. However, in practice, $t_x < 1$, $t_y > 0$, $\varepsilon \neq 0$, therefore has a significant impact on the accuracy of obtaining polarization information.

It can be seen that the inconsistency of the maximum transmittance, minimum transmittance, and transmission axis direction with the theoretical values at different positions on the MPA increases the error in the inversion of the incident light polarization information.

3.     Calibration error.

Currently, MPA calibration is mainly based on theoretical formulas and experiments to calculate the parameters of the first row of the Muller matrix of the micropolarizer. In addition, it calculates the polarization state and Mueller matrix of the light received by the detector to realize the polarization-preserving method of the target degree. The experimental process is broadly as follows: first, the incident light power is set, and then the power remains unchanged; secondly, the polarization characteristics of incident light are modulated by a polarizer, and the values of each parameter in the incident light Stokes vector are considered known. Then, the Mueller matrix's first row parameters are calculated by establishing equations. Then, we replace SP (super pixel) and recalculate. There are four pixels in each SP, so it is necessary to conduct four quaternion equations to calculate all parameters and get all parameters in the first row. Finally, we multiply

the MPA Muller matrix solved by the vector and the Stokes vector of the outgoing light to invert the polarization state of the light before it enters the system. The method of calculating the first-row parameters of the Mueller matrix of a micropolarizer is given in Equation (18) [30].

$$
\begin{cases}
m_{11} I_1^{\mathrm{in}} + m_{12} M_1^{\mathrm{in}} + m_{13} C_1^{\mathrm{in}} + m_{14} S_1^{\mathrm{in}} = I_1^{\mathrm{out}} \\
m_{11} I_2^{\mathrm{in}} + m_{12} M_2^{\mathrm{in}} + m_{13} C_2^{\mathrm{in}} + m_{14} S_2^{\mathrm{in}} = I_2^{\mathrm{out}} \\
m_{11} I_3^{\mathrm{in}} + m_{12} M_3^{\mathrm{in}} + m_{13} C_3^{\mathrm{in}} + m_{14} S_3^{\mathrm{in}} = I_3^{\mathrm{out}} \\
\qquad \cdots
\end{cases}
\tag{18}
$$

2.2.2. Systematic Bias-Preserving Methods

Among the factors of polarization error in MPA-type imaging systems [31], one of the main reasons is the non-uniformity of the system response, so a polarization calibration model based on the camera response value and incident light power is designed. The experiments are designed to acquire the maximum/minimum transmittance of the Mueller matrix and polarization direction at each position in MPA. The purpose of polarization preservation is achieved by the inversion of the outgoing light information and the Mueller matrix.

An SP consists of four micropolarizers in different directions (0°, 45°, 90°, 135°). Equation (19) can be obtained by calculating each micropolarizer once. By solving the equations, the function correspondence of each parameter of the incident Stokes vector with the response value of outgoing light, maximum/minimum transmittance, polarization direction, and camera response value can be obtained. Furthermore, the response value of the camera can be obtained by processing the target image, so the polarization information of the actual incident light can be inverted by determining the actual values of the maximum/minimum transmittance and polarization direction [32].

$$
\begin{cases}
I_0^{\mathrm{out}} = \left( I_0^{\mathrm{in}} + \frac{t_{x0}-t_{y0}}{t_{x0}+t_{y0}} \cos 2\theta_0 \cdot M_0^{\mathrm{in}} + \frac{t_{x0}-t_{y0}}{t_{x0}+t_{y0}} \sin 2\theta_0 \cdot C_0^{\mathrm{in}} \right)/2 \\
I_{45}^{\mathrm{out}} = \left( I_{45}^{\mathrm{in}} + \frac{t_{x45}-t_{y45}}{t_{x45}+t_{y45}} \cos 2\theta_{45} \cdot M_{45}^{\mathrm{in}} + \frac{t_{x45}-t_{y45}}{t_{x45}+t_{y45}} \sin 2\theta_{45} \cdot C_{45}^{\mathrm{in}} \right)/2 \\
I_{90}^{\mathrm{out}} = \left( I_{90}^{\mathrm{in}} + \frac{t_{x90}-t_{y90}}{t_{x90}+t_{y90}} \cos 2\theta_{90} \cdot M_{90}^{\mathrm{in}} + \frac{t_{x90}-t_{y90}}{t_{x90}+t_{y90}} \sin 2\theta_{90} \cdot C_{90}^{\mathrm{in}} \right)/2 \\
I_{135}^{\mathrm{out}} = \left( I_{135}^{\mathrm{in}} + \frac{t_{x135}-t_{y135}}{t_{x135}+t_{y135}} \cos 2\theta_{135} \cdot M_{135}^{\mathrm{in}} + \frac{t_{x135}-t_{y135}}{t_{x135}+t_{y135}} \sin 2\theta_{135} \cdot C_{135}^{\mathrm{in}} \right)/2
\end{cases}
\tag{19}
$$

The experiment of measuring the true transmittance axis of a micropolarizer is designed based on Malus' law. The first step is to open the blackbody radiation source, set the blackbody temperature, and collect the dark current. The second step is to calibrate the transmission axis angle of the rotating polarizer by a polarization state measuring instrument. Thirdly, the initial transmission axis direction of the rotating polarizer is set to 0°, and the image is collected every 1° in the range 0~180°. Fourth, the software extracts the camera response values and saves the response data as a matrix. Finally, taking the angle of the rotating polarizer as the *x*-axis and the camera response as the *y*-axis, the relationship curve is drawn. To overcome the problem that the curve is not smooth due to individual data errors, the least squares method can be used to optimize the curve. The optimized curve is analyzed, and the point with the largest *y*-axis, i.e., the most significant camera response value, is extracted. The horizontal coordinate corresponding to this point is the direction of the transmission axis of the micropolarizer. The direction of the transmission axis of the micropolarizer collected from the experiment is saved as a matrix.

## 3. Experimental Design

### 3.1. System Architecture of the Experimental Setup

The team proposes a general scheme of a multidimensional high-resolution optical imaging system based on an airborne platform for ocean monitoring. The system consists of electric control auxiliary, servo turntable, information processing, spectroscopic, infrared

polarization imaging, and spectral imaging subsystems. The system composition is shown in Figures 2 and 3.

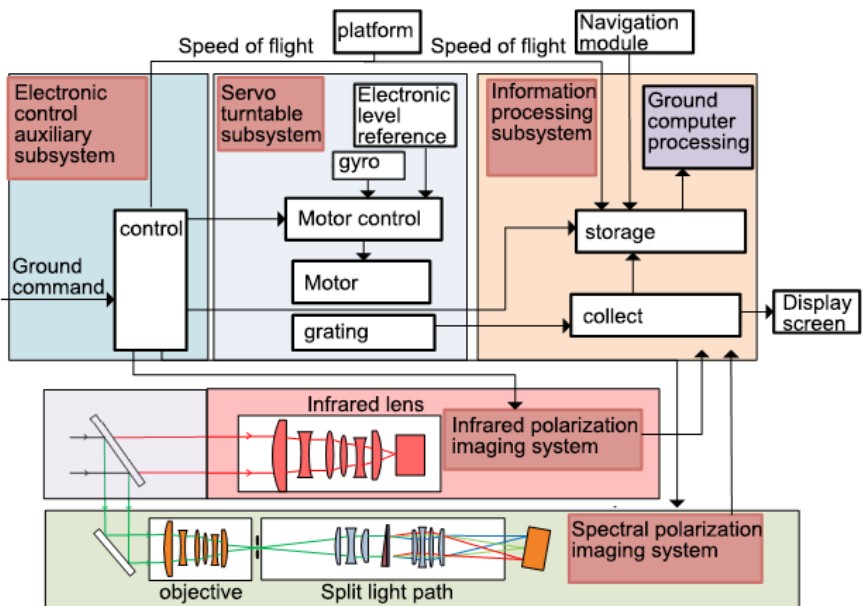

**Figure 2.** The system composition block diagram.

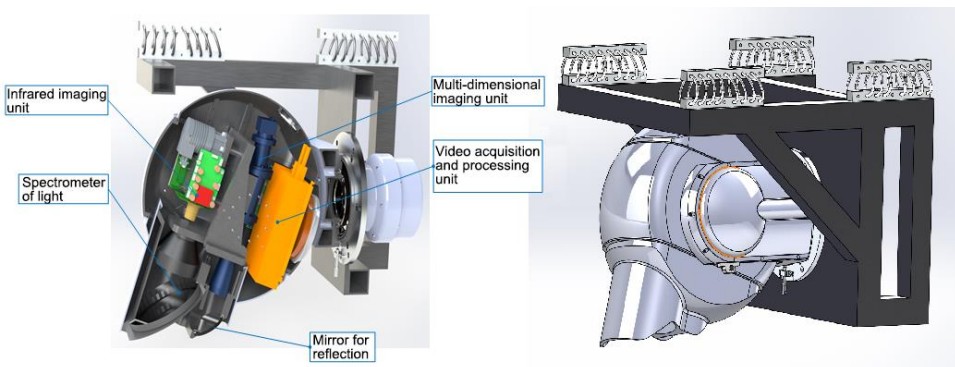

**Figure 3.** Polarization imaging subsystem.

Through testing, the achievable indicators are (i) spatial resolution 0.4 m/pixel@2 km, (ii) spectral resolution better than 10 nm, (iii) specific band 2 nm, (iv) scan area 800 m@2 km, (v) frame rate 104 Hz, (vi) visible polarization unit volume 350 mm × 50 × 50 mm, weight about 5 kg, (vii) infrared unit volume 400 mm × 350 mm × 350 mm, weight is about 33 kg, (viii) NETD is 15 mk, (ix) resolution is 640 × 512, (x) response band is 8~12 μM, (xi) focal length is 75 mm, (xii) F/# is 2.

### 3.2. Infrared Polarization Information Acquisition Solution

The polarization infrared camera uses an uncooled focal plane array polarization detector chip. This chip can simultaneously obtain the polarization information in four directions: 0°, 45°, 90°, and 135°. It can also solve the polarization degree and angle images in real time. In essence, the uncooled focal plane array detector chip uses the direct growth of metal gratings on the image element to integrate the micropolarization array and focal plane. The schematic diagram of the common and polarization array structure is shown in Figure 4.

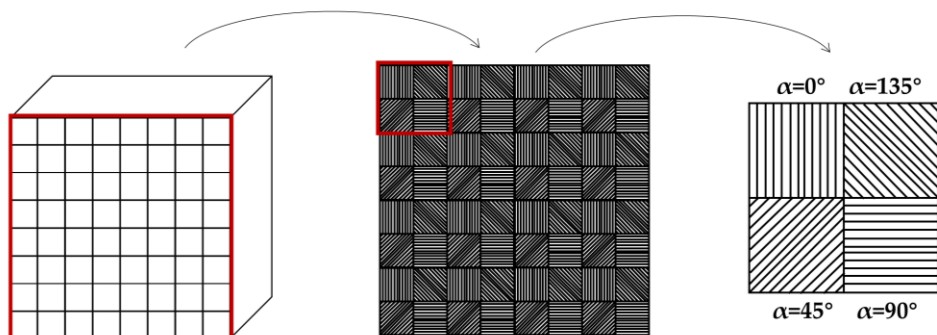

**Figure 4.** Schematic diagram of the copolarized array structure.

The long-wave infrared camera adopts the unrefrigerated infrared MPA detector of North Guangwei Technology INC., and its performance is shown in Table 2.

**Table 2.** The camera performance parameters.

| Parameter | Index |
|---|---|
| Material | Vanadium oxide |
| Specification | 640 × 512 |
| Pixel size | 17 μm |
| Spectral response range | 8–12 μm |
| Operating temperature | −40 °C~60 °C |
| NETD | 80 mK |

In Figure 5, the common focal plane array is a box representing a pixel, and the whole array is composed of the same pixel. Furthermore, the polarization focal plane array, the basic periodic unit, is composed of four images (such as those in the dashed box in the figure), which are not equivalent. Each image represents a polarization direction. The gray line indicates the metal grating, the number in parentheses is the image element number; the first number is the row number, the second number is the column number; the angle below the number (the positive angle with the $x$-axis) indicates the polarization direction, and the polarization direction is perpendicular to the grating direction, e.g., the upper left corner image element (1, 1) 90° means that the image element (located in the first row and the first column) polarization direction is 90°. The other areas of the array are composed of 2 × 2 cells translated into the dashed box.

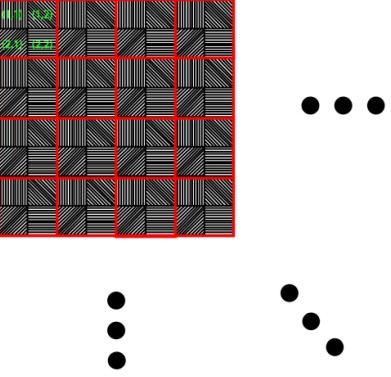

**Figure 5.** Polarization unit translation step (2 a).

### 3.3. Design of the Polarization Information Processing Module

Stokes vectors represent fully polarized, partially polarized, and even natural light. Their components are time averages of light intensity with an intensity scale, which can

be directly detected by detectors. Therefore, imaging polarization detection is mainly described by Stokes vectors.

The Stokes parameter is represented by the light intensity in each polarization direction received by the detector focal plane, so the corresponding light intensity should first be calculated from the detector output voltage $I_\theta$. The detector output voltage $V_{out}$ can be expressed as

$$V_{out} = A + BI_\theta, \tag{20}$$

where A and B are constants and are not calibrated, according to the above equation, $\Delta V_{out} = B \cdot \Delta I_\theta$, this is the starting point for processing polarization data. The Stokes component S0, S1, and S2 images are obtained from $I_\theta$, and the polarization degree images and polarization angle images are further calculated for the subsequent image processing.

Assuming that the original pixel pitch is (a), the $2 \times 2$ polarization cell side length is (2 a). Therefore, the polarization focal plane array is composed of $2 \times 2$ polarization cells with side length (2 a) in translation. According to the translation step, two polarization information processing schemes are as follows.

(1)    The row and column direction translation step size is (2 a):

Starting from the first $2 \times 2$ polarization cell composed of pixels (1, 1), (1 ,2), (2, 1), (2, 2), the polarization focal plane array is divided into $320 \times 256$ $2 \times 2$ cells with no overlap between cells (e.g., in Figure 5, each small red box represents a polarization cell). Next, we calculate the Stokes vector based on the four pixel values of each cell.

The pixel response values of the same transmittance axis in the image are extracted and stored in four matrices, respectively. Each matrix corresponds to an image in the direction of the transmittance axis. At this point, the resolution of the image relative to the original image is halved. After that, interpolation method is used to interpolate the blank position of the matrix by using the known response values in each matrix. In this way, the $0°, 45°, 90°,$ and $135°$ polarization direction images with the same resolution as the original image can be obtained, and the four images have the same field of view. Then, the data at the same position in the four matrices were calculated by Formulas (1) and (2) to obtain the intensity and polarization degree of the micro polarizer at different positions.

(2)    The translation steps in the row, and column directions are (a):

Starting from the first $2 \times 2$ polarization cell composed of pixels (1, 1), (1, 2), (2, 1) (2, 2), the polarization focal plane array is divided into $639 \times 511$ $2 \times 2$ cells with overlap between cells (as in Figure 6, each same-colored box represents a polarization cell) with row direction step (a) and column direction step (a) in the entire focal plane array. Then, Stokes vector is calculated based on the four pixel values of each cell. The image resolution obtained in this way is $639 \times 511$, with almost no loss of resolution compared with the original image.

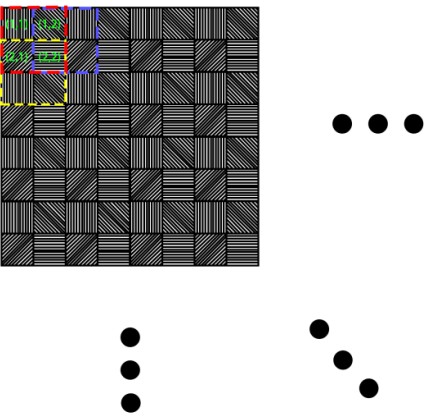

**Figure 6.** Polarization unit translation step (a).

## 4. Bias-Preserving Experimental Validation and Data Analysis

### 4.1. Experimental Scheme of Infrared Polarization Protection

In terms of experimental equipment selection, the image acquisition device used in this study is an infrared MPA camera with a resolution of 640 × 512; the blackbody is a QXHT-X80MYB blackbody furnace with a temperature range of 0~80 °C, a temperature resolution of 0.01 °C, and an effective radiation surface of Φ70, which meets the requirements of the actual working environment. The rotating polarizer is a line polarizer produced by Edmund, with a maximum polarization rate of over 80% and an extinction ratio of 1:1000. To reduce the field of view error, the center of the black body, the center of the rotating polarizer, and the center of the detector should all be located in the same line.

### 4.2. Experimental Results and Analysis

Figure 7 shows the distribution of the transmission axis errors. Again, it can be seen that the error of the polarizer's transmission axis at different positions is different. However, there is no obvious pattern of error variation, and the error range is generally within 15°.

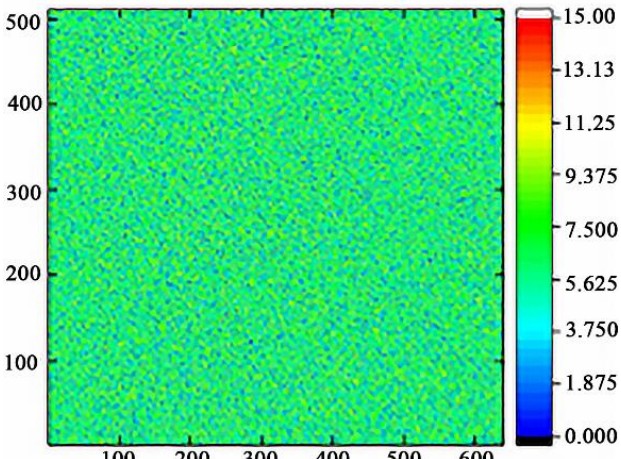

**Figure 7.** Distribution of errors in the direction of light transmission axis.

The transmittance and maximum/minimum transmittance of the micropolarizer were then measured. The blackbody temperature was changed several times without rotating the polarizer, the camera response value data was collected, and the relationship between the response value of each micropolarizer and temperature was determined. In addition, (160, 128), (320, 256), and (480, 384) micropolarizers were selected for curve fitting. It can be seen that the relationship between the camera response value and temperature presents a linear function. The bias maintenance coefficient can be determined by extracting the slope. The Mueller matrix of the entire MPA can be calculated when the direction of the transmittance axis and the comprehensive transmittance of the micropolarizer are known.

The Mueller matrix and camera response values can be used to invert the polarization state of the target, reduce the imaging error, and ensure the accuracy of polarization measurement.

Target comparison before and after correction is shown in Figure 8. It can be seen that the image clarity after correction is significantly higher than that before correction. Details that were blurred or difficult to distinguish before the correction become clear after correction. The contrast of the corrected IR intensity image is improved by more than 3%, and the contrast of the corrected IR polarization image is enhanced by more than 3%. It shows that deflection-preserving can effectively improve the target imaging accuracy.

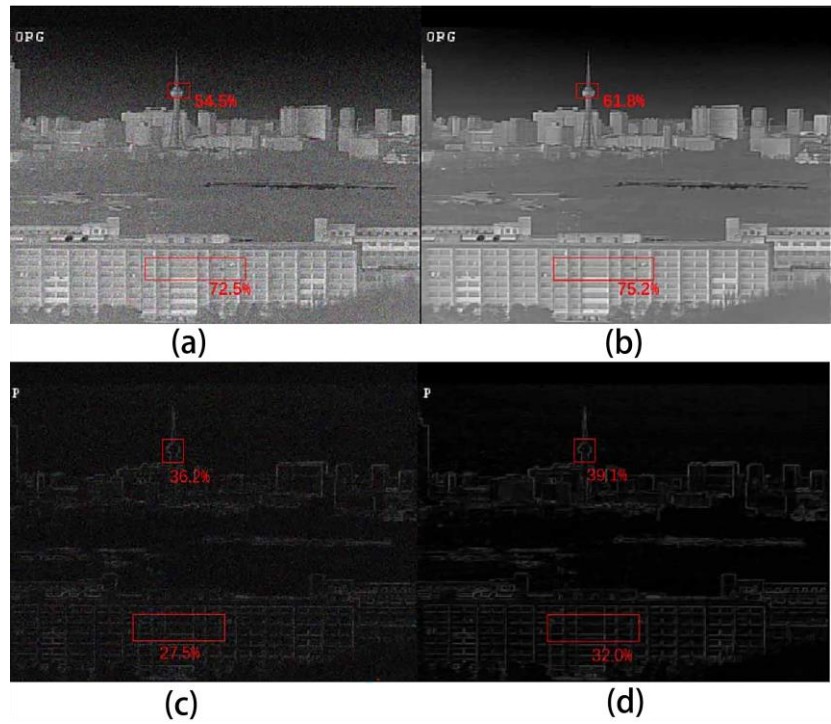

**Figure 8.** Target contrast before and after correction. (**a**) Pre-calibration intensity; (**b**) post-calibration intensity; (**c**) pre-calibration polarization; (**d**) post-calibration polarization.

## 5. Infrared Oil Species Differentiation Experiment

### 5.1. Oilfield Differentiation Experiment Scheme

This experiment aims to measure the intensity and polarization information of different oil spills and analyze the difference of each oil species in details. It also seeks to verify the feasibility of using polarization technology to distinguish marine oil spills and provide support for detecting and managing marine environments. The experimental equipment was prepared prior to the experiment. The image acquisition equipment was selected based on the multi-dimensional high-resolution optical imaging system of the airborne platform for marine monitoring, and the experimental equipment was calibrated to ensure the accuracy of the results. The experimental platform was built on the outdoor site so that the data would be more representative and closer to actual values in the marine environment. Then, the seawater and oil spill were injected into the pool and acrylic tube, and the preparation stage was completed. To investigate the effect of different observation angles on the polarization characteristics of the oil spill, side and top shots of the oil spill were taken under the condition of maintaining a certain angle of incidence; compared to the indoor experiments, the outdoor environment experiences small changes every moment, and these changes can make specific differences in the polarization of the target. Still, the size of the differences and the trend of changes are difficult to quantify. Suppose only one set of images is taken. In that case, it will increase the probability of chance data, and the number of shots will also lead to the deterioration of real-time performance. Therefore, the number of shots for each angle was set to five. After the experiment, the waste oil is disposed of reasonably to avoid environmental pollution. Finally, the acquired image data are processed to eliminate the data with little effect on oil species differentiation, and the oil spill intensity and polarization information are extracted and calculated to analyze the difference in characteristics of different oil species to verify the feasibility of polarization technology to achieve oil spill species differentiation.

The experimental setup mainly consists of an unmanned aircraft, a multidimensional high-resolution optical imaging system for ocean monitoring, a PVC pool, a containment device, an infrared polarization camera, and an electronic computer. A PVC pool of 3 m in length, 2 m in width, and 0.75 m in height was selected as the experimental container.

To prevent the oil spill from spreading, the pool has a containment device made of acrylic with an inner diameter of 15 cm, a wall thickness of 0.5 cm, and a height of 0.62 m. The transparent wall of the tube has a high transmittance and can effectively reduce the interference of stray light from the wall. As the polarization characteristics of the target in the marine environment change constantly, the micropolarizer array detector, which can obtain polarization images in real time, is used to ensure detection accuracy.

Five common marine oil spills, such as fuel oil, palm oil, crude oil, gasoline, and diesel oil, were selected as experimental samples, and seawater was used as the comparison sample. The samples were numbered, and the oil parameters were as detailed in Table 3.

**Table 3.** Oil and seawater parameters.

| Number | Samples | Density (g/mL) | Color |
|:---:|:---:|:---:|:---:|
| 1 | Fuel | 0.821 | Dark blue |
| 2 | Palm oil | 0.836 | Brownish yellow |
| 3 | Crude oil | 0.882 | Black |
| 4 | Seawater | 1.025 | Colorless |
| 5 | Gasoline | 0.737 | Light yellow |
| 6 | Diesel | 0.835 | Light cyan |

The experimental diagram is shown in Figure 9. First, we inject the same height of seawater into the PVC pool and acrylic tube to simulate the marine environment, and then pour the corresponding oil into the acrylic tube until the oil film covers the water surface; at this time, the tube is the oil spill area. We then adjust the observation angle of the device to take side shots of the sample, specify the horizontal direction as 0°, measure the solar altitude angle of 45°, orientation 180°, camera optical axis and horizontal angle of 43°, to obtain five sets of side shots of each sample. Then, we adjusted the observation angle of the device to take a top shot of the sample. The angle between the camera's optical axis and the horizontal was 90° and obtained five sets of 0°, 45°, 90°, and 135° polarization direction images of the visible light and long-wave infrared of the sample and solved the intensity, polarization, and polarization angle images. The experimental demonstration diagram is shown in Figure 10. The experimental effect is shown in Figure 11.

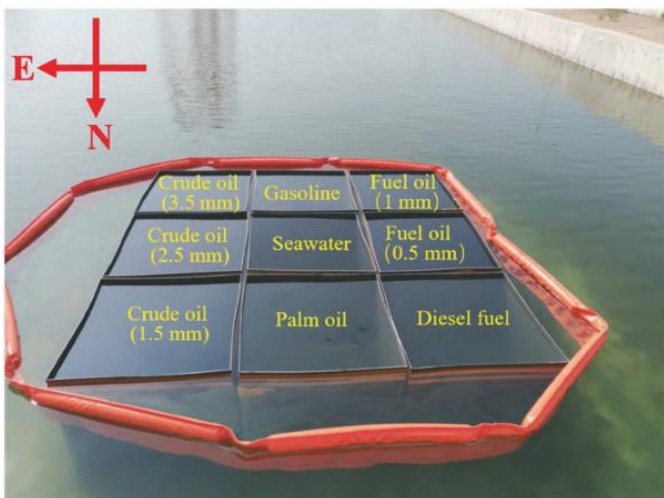

**Figure 9.** Experiment diagram.

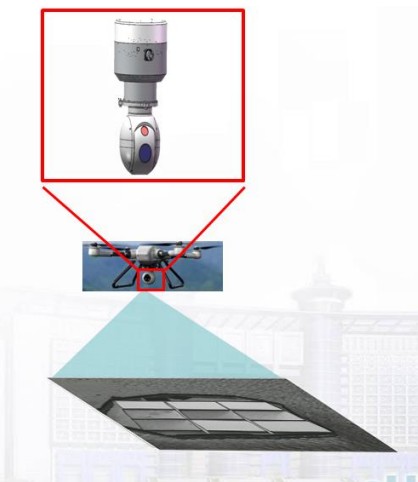

**Figure 10.** Experimental demonstration diagram.

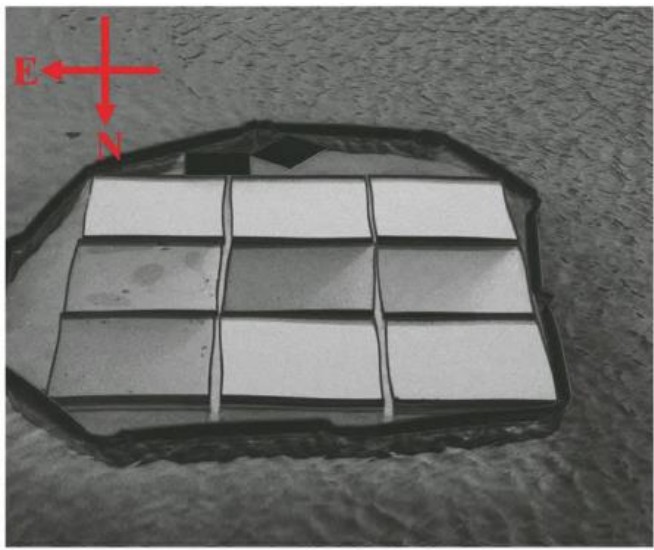

**Figure 11.** Experimental effect diagram.

*5.2. Experimental Results and Analysis*

To ensure that the data are not explainable by chance alone, 95% confidence intervals were calculated for the intensity and polarization data of the five sets of samples:

$$\mu - 1.96 \times \frac{\sigma}{\sqrt{n}}, \mu + 1.96 \times \frac{\sigma}{\sqrt{n}}, \tag{21}$$

where $\mu$ is the mean of the gray pixel value, $\sigma$ is the standard deviation of the gray pixel value, and n is the number of experimental groups. In order to distinguish various oil species, the contrast of each oil species relative to seawater was calculated by taking seawater as the reference and the mean value of the confidence interval as the reference value. The intensity image contrast $C_I$ and polarization image contrast $C_P$ were expressed as

$$C_{\mathrm{I}} = \frac{I_{\mathrm{oil}} - I_{\mathrm{sea}}}{I_{\mathrm{sea}}}, \tag{22}$$

$$C_{\mathrm{P}} = \frac{P_{\mathrm{oil}} - P_{\mathrm{sea}}}{P_{\mathrm{sea}}}, \tag{23}$$

where $C_P$ denotes polarization image contrast; $I_{oil}$ denotes oil spill intensity image grayscale value; $I_{sea}$ denotes seawater intensity image grayscale value; $P_{oil}$ denotes oil spill polarization image grayscale value; and $P_{sea}$ denotes seawater polarization image grayscale value.

From Figure 12 and Table 4, it can be seen that the reflected IR intensity of each sample from the highest to the lowest is crude oil, fuel oil, palm oil, diesel oil, seawater, and gasoline. The 95% confidence intervals for the intensities of gasoline, fuel oil, and palm oil do not intersect and are significantly different. This indicates that the intensity images can distinguish the above three oil spills. The 95% confidence intervals for seawater, diesel oil, and palm oil intensities were less different and even intersected. This indicated that it was difficult to distinguish seawater, diesel, and palm oil using intensity images. The polarization of the reflected light of each sample was from gasoline, fuel oil, diesel oil, palm oil, crude oil, and seawater. The 95% confidence intervals of the corresponding polarization did not intersect, indicating that the polarization can be used to distinguish oil species. The contrast of IR polarization of each oil species relative to seawater is generally higher than the contrast of IR intensity. In contrast, for seawater, diesel oil, and palm oil, which are more difficult to determine in intensity information, their polarization information shows significant differences, further verifying the feasibility of using polarization to distinguish oil spills.

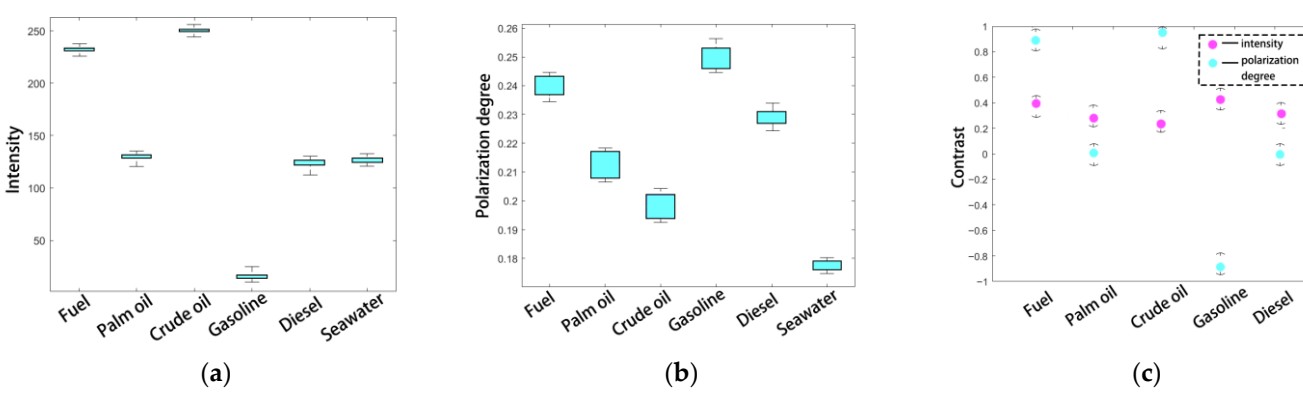

**Figure 12.** Confidence interval for each sample at side shot. (**a**) Infrared intensity; (**b**) infrared polarization; (**c**) comparison of infrared intensity and polarization of each oil species relative to seawater.

**Table 4.** Confidence interval of visible/infrared intensity and polarization for each sample at the side shot.

| Samples | Infrared Intensity | Infrared Polarization |
|---------|--------------------|------------------------|
| Fuel | [231.57, 232.81] | [0.237, 0.243] |
| Palm oil | [128.92, 130.68] | [0.208, 0.217] |
| Crude oil | [249.50, 250.50] | [0.194, 0.202] |
| Gasoline | [14.09, 15.81] | [0.246, 0.253] |
| Diesel | [123.01, 126.29] | [0.227, 0.231] |
| Seawater | [124.82, 128.00] | [0.176, 0.179] |

## 6. Conclusions

Firstly, we carried out research into the current situation of marine oil spill polarization detection technology. To tackle the problem of the low efficiency of oil species identification in the traditional intensity-based detection method, we proposed to combine marine oil spill polarization detection technology with traditional intensity detection techniques. Secondly, a long-wave infrared polarization degree model was constructed. The effects of polarization characteristics of the optical system, MPA processing and manufacturing, detector photoelectric response, and calibration correction on the MPA imaging error were analyzed. The calculation method of the influencing factors in the Mueller matrix of the micropolarizer was studied. The factors that are difficult to measure directly were compre-

hensively calculated to avoid the superposition of errors caused by multiple calculations of parameters, which improved the accuracy of the calculation of the Mueller matrix of the micropolarizer and ensured the accuracy of the inversion of the polarization state of the incident light. Then, a multi-dimensional high-resolution optical imaging system based on an airborne platform for ocean monitoring was designed according to the actual demand. Through simulation analysis, the designed system index met the oil spill detection requirements. Finally, the contrast between the corrected infrared intensity image and the polarization image was improved by more than 3% by designing a polarization-preserving experiment. The polarization detection experiment on a marine oil spill verified that there are apparent differences in the polarization characteristics of different oil spills. Furthermore, the contrast of polarization between oil species was generally higher than 5% and confirmed the polarization's advantages in marine oil spill detection.

The deflection preserving method studied in this paper can only reduce the influence of noise through filtering, so the accuracy of deflection preservation needs to be improved. Moreover, in an actual marine environment, there are many factors affecting the image quality. For example, the energy attenuation of light in the propagation process, the stray light in the sky clouds and the surrounding environment, etc. Due to the limited research time, each factor cannot be analyzed concretely, and could therefore be studied further in future work. In the future, this study can provide a theoretical basis for the development and detection of airborne polarization oil spill instruments, guide the recording of infrared polarization oil spill detection applications, and can also be used in sea search and rescue, island surveillance, red tide monitoring, ship tracking, and other application fields.

**Author Contributions:** Conceptualization, H.S. (Hongyu Sun); methodology, H.S. (Hongyu Sun); software, H.S. (Hongyu Sun); validation, H.S. (Hongyu Sun); formal analysis, Q.F.; investigation, Q.F.; resources, Y.L.; data curation, Y.L.; writing—original draft preparation, H.S. (Haodong Shi); writing—review and editing, L.M.; visualization, Z.L.; supervision, J.L.; project administration, J.W.; funding acquisition, H.J. All authors have read and agreed to the published version of the manuscript.

**Funding:** This research was funded by Natural Science Foundation of China grant number 61890963, 61890960 and 62127813.

**Institutional Review Board Statement:** This study did not involve humans or animals.

**Informed Consent Statement:** This study did not involve humans.

**Data Availability Statement:** Data is not available due to privacy restrictions.

**Conflicts of Interest:** The authors declare no conflict of interest.

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
