# Peer review of "Long-Wave Infrared Polarization-Based Airborne Marine Oil Spill Detection and Identification Technology"

_photonics, doi:10.3390/photonics10050588_

Round 1

Reviewer 1 Report (Previous Reviewer 1)

1.The Introduction Section lacks the literature of relevant research. Is such a literature review the proper expression of an academic paper:

'De Carolis et al. in 2012 used near-infrared MERIS and MODIS images to estimate the thickness of the ocean oil slick [12-16], and Dayi et al. in 2010 used UV push-sweep imaging remote sensing method to monitor the real oil spill pollution on the sea surface [17-21]. Lacava, T. et al. in 2017 used data from the visible channel of the Moderate Resolution Imaging Spectroradiometer (MODIS) on the As a result....'

2.The references are extremely confused. Conference and  Journal papers cannot be distinguished. The same article repeated?

3.English expression needs to be strengthened.

4. In response to Opinion 5 of the first round of review, the revision is more chaotic. How can the author put the experimental description at the beginning of the theoretical section?

5. There are too many mistakes in expression to give examples one by one. The author's carelessness has reached an intolerable point. It is strongly recommended to submit after editing and proofreading the language.

6. Line 353,How can the marine environment be so simple to simulate? This is just an ideal environment experiment.

7. The context of the article is disordered, especially in Chapter 4.

Author Response

It has been modified as required, thank you.

Reviewer 2 Report (New Reviewer)

This manuscript is devoted to the topical environmental topic of pollution control of water surfaces. The authors conducted mathematical modeling and experimental studies.

At the same time, I draw attention to the poor quality of the Manuscript design:

1. Excessive quoting [1-7], [8-11], [12-16], [17-21] it should be divided into several citations of 2-3 references.

2. In lines 87, 156 and many others, remove extraneous entries.

3. Why is part of the text highlighted with a yellow background?

4. Change the name of Table 1, since there are not only oil parameters, but also water.

5. The graphs of Figure 10 need to be increased and their quality improved.

Main questions and comments on the text of the Manuscript:

1. Why connect the points with segments on Figure 10c if the dependence on the abscissa axis is not assumed?

2. Lines 46-55 do not refer to the Introduction. In the Introduction, a comparative analysis of methods for detecting oil spills, etc. should be given. How is the proposed method better than others?

3. The conclusion should be improved. It is necessary to write not what the authors have done, but an analysis of the results they have obtained and further prospects for their application.

Author Response

It has been modified as required, thank you.

Reviewer 3 Report (New Reviewer)

The paper describes the combination of long-wave infrared imaging and polarimetry in order to better detect various oil spills on seawater. 

The authors never describe the origin of their polarimetric camera, but I found a commercial unit available with a technical note provided by the vendor (attached) that describes the same types of capabilities presented by the authors.  So, the authors need to do a better job on how they advanced the state of the art past the attached technical note.  My comments follow based on an objective reading of the paper.

The typesetting in the review document is wildly inconsistent making reading difficult.  In line text shown as superscripts, different fonts, etc..  All of the references to equations seem to be broken in the original Word document.  These should have been caught prior to submitting the document for review after converting to PDF.

All the oils used in this paper are less dense than seawater (Table 1), meaning they will float and create a thin film of oil on top of a seawater substrate.  This makes them thin film systems of the analyte  (oil) on top of an infinite substrate (seawater).  Your model does not address this and how this might affect the signals.  You never define your usage of Stokes vectors nomenclature.  Your system can measure only S0, S1, and S2 since you have no phase measuring elements (waveplates).  Do you define your Stokes vectors as S1, S2, S3, S4?  Different authors use different designations.  This is why you should define your convention.

Line 40:  Please define “YIN”

Line 58:  Define long wave infrared.  I know what that means to me, but it may differ for other readers.

Line 66:  Create figure showing the relationship between the entities in Eq. 1.

Line 67:  it will be less painful for the reader and clearer if the equation terms are defined in a table

Line 70:  the index of refraction of both the oils and seawater is complex and should be noted as it is not strictly real-valued

Line 79:  I would use epsilon in place of alpha and call this emissivity

Line 81:  emittance (what you call alpha) is a function of wavelength and is not a constant for many materials

Line 88:  please provide a reference for the Fresnel Equations and the form the authors’ have chosen. 

You state in lines 91 through 98 that you will build a polarization model based on an ideal sea surface and rough sea surface, and then you don’t.  Except for modeling errors due to imperfections in the polarization elements I don’t see any modeling for the basis for the detection method or the influence of smooth versus rough seawater.

Provide a reference for Equation 10

Equations 8 and 9:  These are odd formulations of the Fresnel equations.  What is the definition of a and b parameters in the equations?  These are not in the text.

Line 102:  MPA never defined prior to use of the acronym.  I assume this stand for Micro-Polarizer Array.

Line 124:  you never describe the basis of the operation of your micropolarizers.  They look like metal grid type polarizers.  This could help the reader understand what errors in manufacturing alter the extinction ratio of each element.

Wavelength sensitivity range of the camera is never specified

Where did Eq. 10 come from?  Reference?

Eq 11:  Should the subscripted be “ik” instead of “tk?”

Line 151:  Define the term SP. Stokes pixel?

Line 211:  is this a microbolometer system?  Is the polarization focal plane array a commercial product?  If so, you should provide the manufacturer information.

Line 242:  shouldn’t the differential of Eq. 15 be delta V = B*delta I?

Line 320:  incomplete sentence:  “…effect of different observation angles on the In order to investigate…”

Line 355 through Line 360:  here a diagram makes more sense to show the angular relationships.

Fig 3.  You need to provide a figure just showing one super pixel with the 4 sub pixels of each metal grid polarizer.

Eq. 18:  did you define what you mean by oil Polarization image?  You obtained information to form 3 Stokes images, did one provide better detection, like S2 or S3?

Line 407:  “And the effects of zenith angle, azimuth angle, refractive index and other factors on the polarization degree of oil spill are analyzed by simulation.”  I don’t see any simulation results in the paper.

Author Response

It has been modified as required, thank you.

Round 2

Reviewer 1 Report (Previous Reviewer 1)

The author should carefully examine the expression of the text.The reading of the article is very obscure.

(1) The parameters explanation of formula (1) is unreasonable, how can 0, 1, 2... be explained as variables.

In addition, all formula arrangements should be revised

(2) Is Figure 1 an explanation of the elements of Formula 1?

(3) "Usually, the fewer the number of refractions, the smaller the change of polarization, the smaller the difference between the angle of the incident light and the outgoing light, the smaller the change of polarization, and the better the optical system preserves polarization." This sentence should be revised as:

Generally, fewer number of refractions ***(leads to, causes) smaller change  of polarization...

(4) "first,...and then...secondly,...Then,...Then,..Finally,..." Have similar sentence structures really been edited and proofread.

(5) Please confirm if Figure 2 is their original image. This blurry image may be a replicated copy.

(6) Is Figure 4 drawn reasonably? I need to read the text information in detail to understand the simple content expressed in Figure 4

(7)"Furthermore, the polarization focal plane array, the basic periodic unit, is composed of four images (sub-images?)"

(8) If authors want the results to be accurate and reliable, I suggest adding infrared images collected from different angles and analyzing them. A single infrared image (Figure 9) and a table (Table 4) are difficult to support the reliability of the results. Especially for comparative experiments without looks like targets.

Author Response

Reviewer 3 Report (New Reviewer)

I apppreciate the authors' considering my comments and incorporating them in the revised version of the manuscript.  It is much improved and will help readers to understand their methods and results better.  It still needs a little text polishing but otherwise is a much better read after corrections.

Author Response

It has continued to be embellished. Now it reads better.

Round 3

Reviewer 1 Report (Previous Reviewer 1)

The paper has basically reached the acceptance level.

Figures 5 and 6 are still unclear, and optimization is recommended before acceptance.

This manuscript is a resubmission of an earlier submission. The following is a list of the peer review reports and author responses from that submission.

Round 1

Reviewer 1 Report

It is of great significance to study the detection and identification methods of offshore oil film based on airborne infrared remote sensing technology. However, the presentation method of the article needs significant improvement, such as:

1. There are many mistakes in the paper.

2. Some of the formula format does not also fully conforms to the specification.

3. Figure 1: If it is not original, give the reference.

4. Figure 6 is so unclear. Is it still meaningful? The mark label information in Figures 7 and 8 is still the same.

5. Is Section 1.2 a general infrared deflection protection study or a specific method for oil spill? It should be stated at the beginning.

6. Should 3.1 and 4.1 be the same title?

Reviewer 2 Report

This is a great research manuscript exploring the infrared polarization detection method of the oil spill. In the manuscript, the author well explains the basic theory of detection together with the detailed error analysis. Then an experimental method is developed, with the real oil detection, the conclusion is believable that some oil has significant difference compare with the seawater in polarization signal. I agree that this is a pretty detailed and meaningful research paper. However, before I agree to publish this manuscript, there are some issues related to the format and pictures that the author needs to correct.

For example, in the error analysis section, there are some letters and words with the different format. In the Figure 1, all words are flipped, not readable. In the Figure 2, the format of description is not constant. There are some other error/mismatches in other paragraphs need to be corrected.